# The Dialogic Health Systems Research Framework (DHSRF): A tool for facilitating self-criticality, researcher interactions and knowledge management in Health Systems Research & Policy Studies

**Ritu Priya[1]\***, **Sayan Das[1,2]**, **Liz M. Kuriakose[1]**, **Madhurima Shukla[3]**, **Amitabha Sarkar[4]**, **Neha Dumka[2]**, **Erin Hannah[2]**, **Nisha Basheer[2]**, **Atul Kotwal[2]**

**1** Centre of Social Medicine and Community Health, Jawaharlal Nehru University, New Delhi, India, **2** National Health Systems Resource Centre (NHSRC), New Delhi, India, **3** Centre for Health and Social Sciences, School of Health Systems Studies, Tata Institute of Social Sciences, Mumbai, India, **4** Health Sciences Unit, Tampere University, Finland and the United Nations Research Institute for Social Development, Geneva, Switzerland,

\* ritupriyajnu@gmail.com

## Abstract

Health Systems Research (HSR) has witnessed significant progress in theory, methodology and practice over the last two decades. The complexity of health systems has allowed for diverse lenses for HSR. However, given the absence of dialogue between the different streams of HSR, the diversity of this field, perhaps its greatest strength, is turning out to be quite the challenge. Without a common language that enables researcher interaction and critical examination of the field, diversity can easily turn into a din. To overcome this confusion, evidence-based policy making requires tools that can assess the diverse evidence generated for designing systems coherent with the desired values and principles. Hence, we need a common research framework for HSR, to understand, describe and explain the systems' structure and functioning, including observed and projected processes of change, across streams. It should be able to make sense of the formulation of HSR, allowing diverse research paradigms their appropriate place in HSR, and make them talk to each other rather than against each other. This paper presents the Dialogic Health Systems Research Framework (DHSRF), developed through a multi-method approach which was theory-based, iterative and reflexive involving an initial systematic narrative review and a later national expert consultation for feedback and validation. The DHSRF locates itself within a public health disciplinary frame and draws upon the comprehensive primary health care principles and a complexity lens. This led to adopting a comprehensive socio-cultural approach to health systems, incorporating their formal and informal components, addressing the techno-managerial and social-economic-political-cultural aspects with understandings drawn from epidemiology, historical

**Data availability statement:** All relevant data are within the paper and its Supporting Information files.

**Funding:** This paper is an outcome of a collaborative project titled "A Review of 'Health Systems Research' for Strengthening Future Knowledge Management, Planning and Policy in India: Mapping State of the Art from Global to Local" between the Centre of Social Medicine and Community Health, Jawaharlal Nehru University and the National Health Systems Resource Centre (NHSRC), MoHFW, Government of India. Most of the researchers have been salaried employees of one of the two institutions, except SD, LK, MS, AS, who received a contractual payment for their work on the project which was located at JNU. This has been from internal funds under the Knowledge Management Division of the NHSRC.

**Competing interests:** The authors have declared that no competing interests exist.

analysis, knowledge pluralism and a bottom-up approach, making the framework context-sensitive, open to diverse perspectives, adaptable to different settings, value-critical and dialogic. It potentially contributes to knowledge management by a) allowing for a comprehensive review of HSR at the proposal and design stage to ensure a 'good fit for purpose', and b) assessing the strength of the outcomes of HSR in relation to the purpose and objective(s) of the research. By surfacing the inner workings of HSR, the framework can galvanise dialogue and debate to enrich the field and facilitate utilisation of its outcomes for policy, planning and implementation.

## Introduction

Health systems are expected to play a crucial organisational role in linking people with appropriate health care while simultaneously addressing the broader eco-social determinants of health in pursuit of health and well-being for all. Health Systems Research (HSR) encompasses the whole spectrum, from individual and community-level experiences of health to health services implementation, planning and policy-making at the national/global scale with everything in between that promotes, restores and maintains health.

The heightened interest in HSR over the past two decades has led to an increasing quantum of research with important conceptual and methodological advances. Given that systems thinking seeks to examine and address systemic problems in their complexity, this expanding interest in HSR should bring positive outcomes for Public Health. At the same time, with this expectation, significant criticism exists that current Health Systems Research (HSR) needs to adopt more comprehensive perspectives and generate sustainable solutions to both existing and emerging health challenges.

### Defining HSR

Over the years, HSR has been diversely defined with relative prioritisation of different constituting elements. For instance, the Alliance for Health Policy and Systems Research (AHPSR), gaining prominence in the decade of the 2000s, puts Health Policy as the central objective of HSR and calls their proposed iteration of HSR, Health Policy and Systems Research (HPSR). [1] However, we tend to agree more with Hoffman et al [2] and consider Policy Studies (PS) to be one among the many components of HSR, with implementation research, evaluation research, and the socio-cultural dimensions of health care as other major components.

In our view, HSR and PS are two closely related and often overlapping, yet distinct interdisciplinary fields with widely differing methodological imperatives. Epidemiology, HSR, Policy Studies, and the exploration of the dynamic SPEC (socio-political-economic-cultural) contexts, represent the essential pillars of public health research. The field of HSR draws from all the others, depending on the purpose of the research and research questions. HSR, including PS, is essentially meant to give a deeper understanding of health systems and their dynamics, which also generates the

evidence for policy making, planning and implementation. Policy studies and the process of policy formulation using HSR evidence need to be distinguished as two separate activities, each with their own relevance and imperatives. We would argue that the socio-cultural and informal components tend to be ignored when contributing to policy formulation becomes the primary purpose of HSR. Therefore, we prefer to use just HSR, with the understanding that PS will integrally be a part of it when required by the objectives and research questions. (See S2 File for more details)

## Conceptualising Health Systems and Research Problems

Scholars find that the limitations of HSR often stem from a lack of coherence between research objectives, conceptualisation of health systems and their effective strengthening, and methodological approaches. [3] The way the health system is visualised in HSR thus becomes crucial since any process of conceptualisation/framing/modelling/defining involves representing the health system issue/research problem symbolically to state assumptions about its very nature. Once the problem definition is set, the rest of the research is typically about amassing further information about the nature of the health systems issue within the remit of its definition. Therefore, what is known about any given problem is determined by how the problem was defined. [4] For HSR, that would primarily be the conceptualisation of the health systems as relevant to the research problem and research questions.

The origins of HSR in public health are from bio-medical, demographic and management disciplines with their classical positivist paradigm. Despite the predominance of the positivist paradigm, from the 1970s onward, HSR was extended beyond its classical roots by a comprehensive public health perspective adopting a pragmatist approach. It incorporated the techno-managerial and the social, political, biological and ecological dimensions of Health Systems for problem solving and designing health service systems and disease control programmes. [5–6] In recent times, a more theory-based approach to the social dimensions of HSR has been highlighted by development economists and others, often distancing it from the core of public health such as epidemiology.

Much literature appeared from various quarters to frame the field of HSR, especially at the beginning of the second decade of this century. The AHPSR, in its HPSR methodology reader published in 2012, listed a lack of rigour and a weak basis for generalisation of findings and thus limited policy application, as the major criticisms against the growing field of HSR. [1] A set of three articles, published parallelly in PLOS Medicine cautioned against the looming threat of disciplinary capture of the nascent field by dominant health research traditions such as epidemiology, biomedicine and clinical research anchored primarily in a positivist paradigm. [7–9] Such positivist framing (with its claim of value-neutrality) arguably reduces health systems to mere vehicles of technological solutions and undermines the role of socio-political context and power differentials that shape them. [7] They proposed the role of social sciences and a realist paradigm as central to HPSR. While this has added immensely to deepening the inquiry of social dimensions of the formal component of the health system, it has tended to fragment the comprehensive public health approach to health systems. [1]

Another growing stream of HSR is that of mathematical modelling and Randomised Control Trials (RCT) for health system evaluations or proposed service interventions. [10] These draw from disciplines such as development economics, management studies and Big Data analysis which tend to be as reductionist as the biomedical approaches. Clearly, one stream views health systems as social institutions and the other as primarily techno-managerial systems.

Accordingly, different configurations of Health Systems exist in HSR, variously prioritising the environmental, social, economic, epidemiological, cultural and institutional dynamics shaping the health system or the more dominant techno-managerial programmes and medical/health services focused organisational approaches. The social view is often deemed too difficult to apply in research practice due to its complex conceptualisation, researcher capacities and time requirements while the techno-managerial view is often criticised for being rather narrow in understanding and scope, thereby, providing limited effectiveness of systems design and policy options. For our HSR review and the framework being presented in this paper, we located ourselves in the comprehensive public health discipline and incorporated a theory-based approach to HSR.

## The Need for a Health Systems Research Framework

Despite the diverse spectrum of approaches adopted for HSR, few studies have reviewed such developments in HSR (since 2010–2012), with none from India. Therefore, identifying the various knowledge paradigms used in HSR, analysing their conceptual and methodological approaches, and reviewing their strengths and limitations in contributing to the techno-managerial as well as SPEC dimensions of health systems strengthening are essential for further deepening the science of HSR and its inputs for policy and practice. With these objectives, the research project, "*A Review of* '*Health Systems Research*' *for Strengthening Future Knowledge Management, Planning and Policy in India: Mapping State of the Art from Global to Local",* was initiated jointly by the Centre of Social Medicine and Community Health, Jawaharlal Nehru University, India and the National Health Systems Resource Centre of the Ministry of Health and Family Welfare, Government of India. Given the dearth of systematic assessments of the nature and content of HSR approaches, this research project sought to map the broad approaches and streams of HSR—conceptual and methodological— through a review of the field. A Technical Advisory Group (TAG) was constituted for an ongoing peer review process and to provide overall support and technical inputs to the project.

The complexity of the health system extends to the field of its research as well, with definitional differences around key terminologies, diverse framings and research paradigms, lack of clarity about the role of values and principles and a dearth of dialogue about the nature and content of evidence generated by HSR. The fit between the different steps in a HSR study, from deciding the purpose of the research and its objectives, formulating the design and operationalisation, to analysing findings and finally recommendations, i.e., the research coherence is seldom assessed. Hoffman et.al [2] call it a 'Standards challenge'. Further, they note a community diversity challenge of HSR. Given the diverse locations, traditions, training and disciplinary moorings of health system researchers who prefer different questions and methods, "there is little that binds health systems researchers together and much that pulls them apart.". They posit that such diversity, which is perhaps the greatest strength of HSR essentially needs a platform to build on which can offer a common language for the use of health systems researchers.

We submit that besides a common language, we need a common research framework that helps in addressing the complexity of the health system, i.e., helps to understand, describe and explain the systems' structure and functioning, including processes of change. The common language and framework should be such that it is able to make sense of the formulation of HSR, allowing the diverse paradigms their appropriate place in HSR, and make them talk to each other rather than against each other.

A research framework makes explicit the underlying and often not-so-clearly stated values, principles, concepts and design elements that shape the research and its outcomes. It supports the researcher in making appropriate design choices and enables reviewers to more effectively assess the strengths and limitations of the study, as well as its evidence-based recommendations. Therefore, given the diversity of contexts and perspectives, a common framework can unpack the HSR: its conceptualisations, methodological approaches and most importantly, its value positions. Such a Health Systems Research Framework (HSRF) will facilitate more holistic reviews, contribute to strengthening the field of research and knowledge management, and enable appropriate utilisation of study findings for health policy, planning and implementation.

Hence, while undertaking the review of HSR studies, there was a felt need to create a research framework. It was thought that an HSRF (as distinct from a HS framework) can contribute to knowledge management in HSR by: a) allowing for a comprehensive review of HSR at the proposal and design stage to ensure a 'good fit for purpose', and b) assessing the strength of the outcomes of HSR in relation to the purpose and objective(s) of the research at the end stage of analysing findings and proposing recommendations. Using a research framework to guide the review of a study design or its outputs is advisable because it will ensure self-reflection, rigour and transparency in the formulation and practice of HSR and encourage informed conversations. This will help the discipline move forward with openness and a better understanding of the evidence generated by HSR and its value in decisions about health systems. Therefore, we call the HSRF we developed, the Dialogic Health Systems Research Framework (DHSRF).

## Research methodology

This paper presents the Dialogic Health Systems Research Framework (DHSRF), developed using a multi-methods approach. The research tool development was undertaken by adopting a theory-based, iterative and reflexive methodology involving an initial formative systematic narrative review and a national expert consultation for feedback and validation. The DHSRF locates itself within a public health disciplinary frame and draws upon the comprehensive primary health care principles and a complexity lens. The tacit knowledge and perspective of the researchers and the expert consultation contributed to making the framework open to diverse perspectives, context-sensitive and yet adaptable to different settings, value-critical and dialogic.

For reviewing HSR papers, our study began with iteratively drafting an analytical HSRF (HSRF-0), as we could not find any standard framework in literature for reviewing HSR. The Complex Adaptive Systems lens was used to conceptualise health system structures, their interlinkages and dynamic processes. [11] These attributes are also inherent in the spirit of the PHC approach and its operational requirements. [12] The initial analytical framework (HSRF-0) was thus developed by modifying the basic conceptual elements of health systems with attributes drawn from the scholarship around the spirit of the PHC approach and understanding of complex adaptive systems (See S1 File for HSRF-0).

However, in the process of the review, we felt the need to strengthen the analytical framework further before it could be applied to HSR papers. Hence, a systematic narrative review was conducted on HSR from 2000 onwards with the keyword "Health Systems Framework" to understand how Health Systems were variously defined and conceptualised in HSR literature (further details about the systematic narrative review can be found in the S1 File). This enabled an examination of the conceptual and methodological contours of health system frameworks available in existing literature.

To develop a Health Systems *Research* (HSR) framework, some elements of HS emerging from the systematic review were added into the version-0 of the HSRF. It was further iteratively modified through the steps of the research process to finally develop the DHSRF as indicated in Table 1 (and each version shared in S1 File), incorporating attributes gleaned from the following three steps.

I. A systematic narrative review of international literature for health systems frameworks was undertaken to generate HSRF version-1 (HSRF-1). The systematic review and the HSRF-1 were presented to the project's Technical Advisory Group (TAG) and their inputs supported the continuing work.

II. This HSRF-1 was used to review a set of purposively selected studies across selected HSR themes in India, which helped to further adapt it in response to the need for capturing relevant attributes across a range of HSR studies. This process generated version-2 (HSRF-2).

III. Finally, HSRF-2 was presented to a group of eminent HS researchers in a national consultation organised under the aegis of this research project, whose detailed feedback was taken to modify the Framework and finalise it for the present, i.e., develop the version-3 or DHSRF.

TAG meetings were held at the end of each phase to share progress and get feedback. The objectives of the National Consultation were communicated in writing to the HS researchers at the time of invitation with the explicit purpose of obtaining their feedback and input on the HSRF. At the Consultation, verbal consent was taken publicly for audio and video recording of the proceedings, which was documented by rapporteurs and witnessed by all present, including the TAG members in attendance. The framework was presented in detail on the first day, and the participants actively engaged with its conceptual and methodological issues. Minutes of the very rich discussion were comprehensively prepared overnight and presented on the second day for modification/corrections required and verbal consent for final approval. The inputs of the TAG and the National Consultation were extremely useful in fine-tuning the Framework.

This paper presents the process and structure of the DHSRF. It is hoped that wider use of this DHSRF will contribute to its further refinement.

**Table 1. Development of the Health Systems Research Framework (HSRF) through different versions.**

| | |
|---|---|
| **HSRF-0** | **• The initial analytical framework generated by incorporating health systems attributes from the PHC and CAS approach in a basic systems research framework.**<br>**• This version was used for reviewing the 60 selected studies in a Systematic Narrative Review of international literature available since 2000.** |
| HSRF-1 | • Developed by incorporating findings from the Systematic Narrative Review:<br> A number of values and principles were added as an additional component of analysis as it emerged that values and principles often determined the different paths taken by research informed by the same paradigm. The values and principles were assessed to see how broadly or narrowly they were conceptualised or operationalised across different components of the HSR.<br>• This version was applied to review Indian HSR studies selected across specific themes as a formative exploration. The themes chosen were-<br>1. Health Systems Research integrated in National Policy and Planning processes,<br>2. A major national health system strengthening intervention (the National Rural Health Mission),<br>3. A specific national health program and its components (the Immunisation Programme),<br>4. Health workforce<br>5. Primary Health Care |
| HSRF-2 | • Developed by incorporating the findings from the formative exploration and review process-<br>The different analytical findings needed to be brought together to understand the value of the research as a whole. Thus, to synthesise the findings of analysis, a Research Coherence framework was prepared to see how different research components fit with each other: Coherence between the study's objectives, the conceptualisation of the problem and the theoretical framework adopted; the objectives, theoretical framework and the study design; the design and data analysis framework; the data and its interpretation; findings and recommendations. |
| The Dialogic Health Systems Research Framework (DHSRF) | • Developed by incorporating suggestions from the national HSR experts' consultation on the HSRF-2.<br>1. The language and presentation were made more reader-friendly for an audience beyond HS researchers, especially policy-makers.<br>2. A glossary was added defining each of the concepts used in the DHSRF (See the S3 File). |

## The Dialogic Health Systems Research Framework (DHSRF)

The DHSRF broadly surfaces the values and principles, the research paradigm and various components that inform the conceptual and methodological framing of any HSR but are seldom stated explicitly. This makes the configuration of HSR transparent for any researcher or reviewer by drawing attention to the various components for consideration and providing a 'common language' accessible to every health system researcher.

The values and principles underlie every step of HSR including the choice of research paradigm. Starting from the statement of objectives and framing of the research questions to the conceptualisation and methodological approach of the research, an interaction of values and principles and the paradigm is evident. The AHPSR methodology reader says that different paradigms lead to addressing different kinds of research questions. Our research, however, found that the same research questions are being addressed by the different paradigms. However, the conceptualisation detailing the research questions and the methodological design differs across the research paradigms adopted (the example of designing of HSR using the framework in the S5 File gives an illustration). Within the same paradigm, studies reveal conceptual and methodological variations based on differing values and principles. Hence, we find that values and principles and research paradigms become the overarching determinants and signifiers of the path taken by any HSR.

With these overarching conceptual attributes, any research must attempt coherence in its conceptualisation of the HS as relevant to the research problem, its research methodology, interpretation of data for the findings and recommendations for HS development. In order to undertake the analysis of each of these and then their coherence, a set of three analytical tools were developed. The following (Figs 1, 2, 3) presents the graphical depictions of the three analytical steps using the tools so developed.

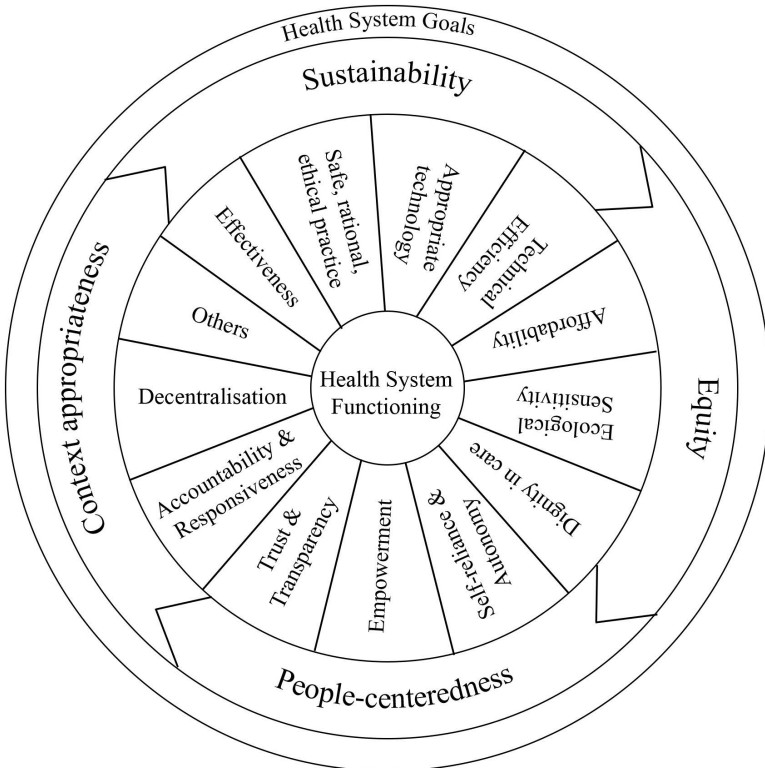

**Fig 1. Graphical Depiction of Values and Principles applicable to Health Systems.**

Values and principles influence each step of research yet most often remain unstated or unexplained. The DHSRF therefore as the first analytical step involves an assessment of the values and principles as applied in the HSR under review. From the public health and health systems literature and our own assessment, we have classified them across two broad categories of Health System Goals and Health System Functioning. Health systems strengthening, people centredness, equity and sustainability are central to public health relevant health systems goals.

Analysis of health system conceptualisation is the next step. The conceptualisation follows the typical systems analysis of boundary and subsystems. In the subsystems, we stress on including both structures and processes. One of the two additions that we make to the health systems conceptualisation is the Dynamic Elements of Health System and its Context (Ecosystems, Socio-political contexts of health care, Meaning systems of health care and Informal societal arrangements for health) which, irrespective of where the boundary of the health system is drawn, influence the subsystems and their functioning in diverse combinations. Each of these has been detailed in the glossary (S3 File).

The other addition is of the Health Systems Vantage Point. A health system vantage point can be either bottom-up (community perspective) or top-down (institutional), or a combination of the two. Deciphering this for an HSR study involves analysis of the perspective underlying its HS framework, the framing of research questions and methods of health systems research with special reference to how the relationship between the community and the institutional dimensions of the health system are conceptualised. How far do the objectives and ways of achieving them, the analytical frame and recommendations derived from it for implementation, planning and policymaking include the community's agency and real-life context.

The third step involves the analysis of methodological approaches which includes the overarching research paradigm, the nature of disciplinary interaction (multidisciplinary, interdisciplinary, transdisciplinary), the analytical approaches

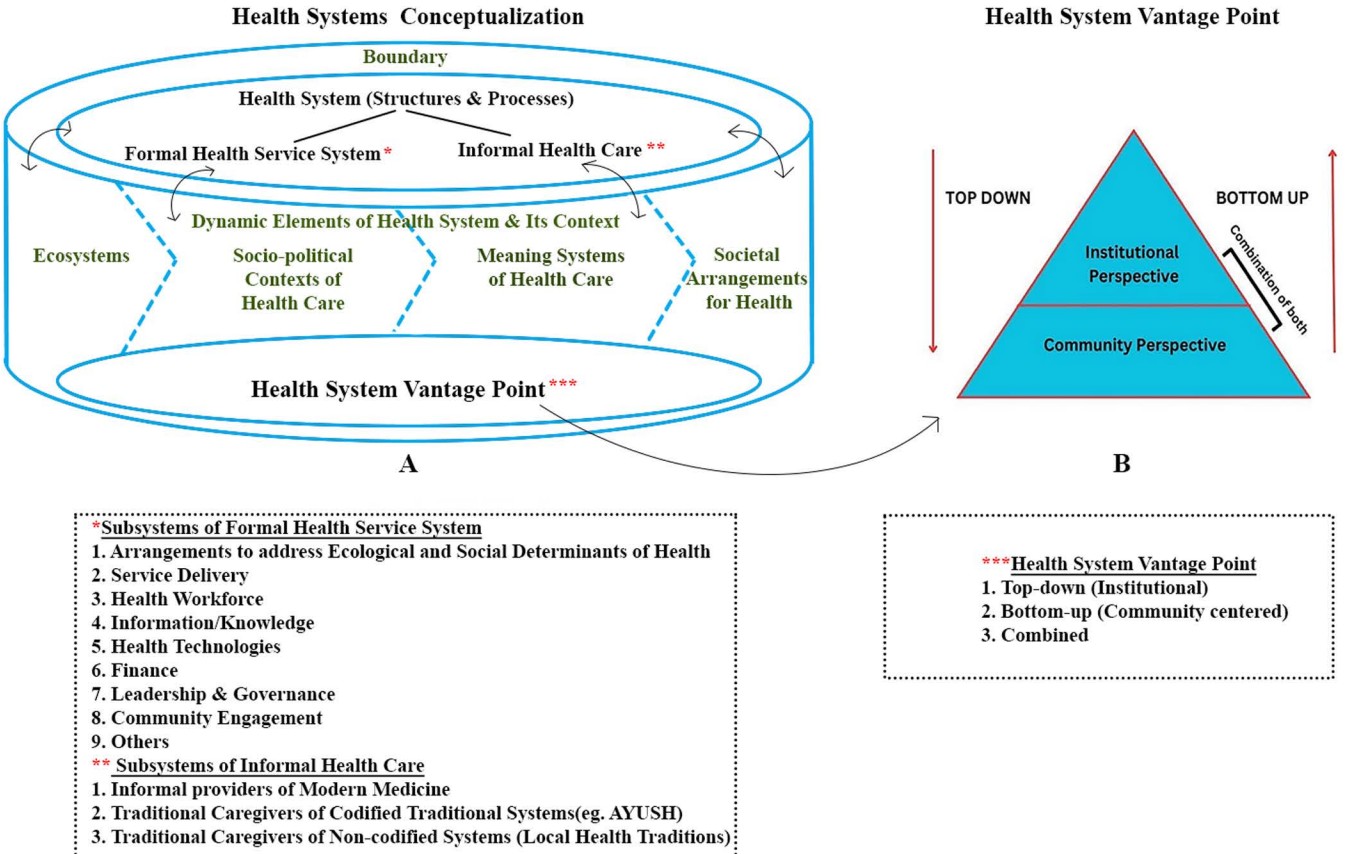

**Fig 2. Graphical depiction of health system conceptualisation and Health Systems Vantage Point.**

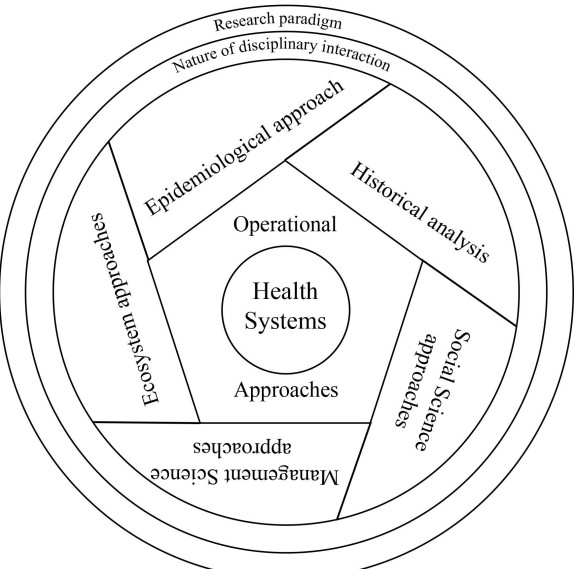

**Fig 3. Graphical Depiction of Methodological Approaches in Health Systems Research (a lens focused on examining the Health System).**

(Epidemiological approach, Historical analysis, Social Science approaches, Management science approaches and Ecosystem approaches) and the operational approach (various intervention or non-intervention study designs). Quality of operationalisation of the research is not built into the framework and should be assessed using any existing tool relevant to the nature of study.

While the three analytical tools help unravel different strands of underlying values and principles, HSR conceptualisation and methodology, they are brought together in relation to each other through the assessment of research coherence. Defined as, "the fit between the aim, the philosophical perspective adopted, and the researcher role in the study as well as the methods of investigation, analysis and evaluation undertaken by the researcher", [13] research coherence is perhaps the most crucial measure of the rigour and validity of research (Fig 4).

## The DHSRF structure

The Dialogic Health Systems Research Framework has two components:

A.   An initial description of the research, and

B.   An assessment of research coherence

### A. Description of the research

| | |
|---|---|
| **Name of study/Title of report/paper** | |
| **Year of publication** | |
| **Author/s** | |
| **Institutions** | |
| **Funding** | |
| **Type of Institution** | |
| **Type of Research**<br>**(Implementation/Evaluation/Policy Analysis/ Policy Studies//Perspective/**<br>**Review/ Synthesising Research/Others)** | |
| **Objective of the study** | |
| **Methodology of the study** | |
| **Level of analysis (Global/Inter-state/National/State/District)** | |
| **Area of analysis (Urban/rural/peri-urban/All)** | |
| **Policy implications** | |

### B. Assessment of Research Coherence (synthesised from analysis by the three tools)

| Health Systems Research Components | Research Coherence | Research Paradigm<br>• Conventional Comprehensive Public Health/ Pragmatist<br>• Positivist<br>• Realist closer to Positivist<br>• Realist<br>• Realist closer to Holist<br>• Holist | Strengths of research |
|---|---|---|---|
| Research Objective(s) | | | |
| Research Question(s) | | | |
| Values & principles for the Health System | | | |
| Health System Research conceptualisation | | | |
| Methodological approach | | | |
| Findings | | | |
| Recommendations | | | |

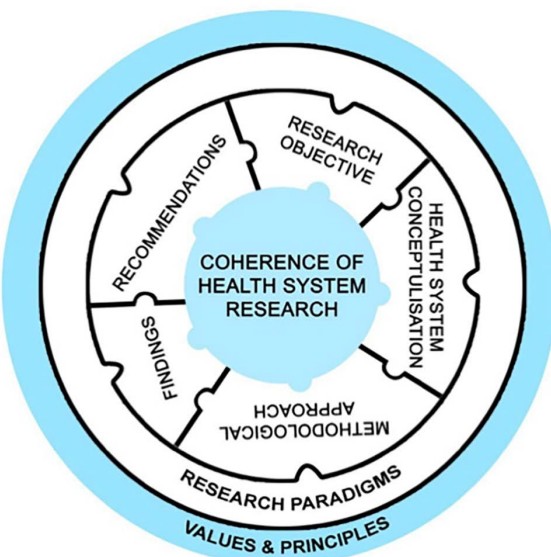

**Fig 4. Graphical Depiction of Research Coherence as the critical basis for assessment of HSR by the DHSRF.**

## Tools for assessment of research coherence

The assessment of research coherence is done through three steps of backroom analysis:

1 Analysis of values and principles for the health system

2 Analysis of health system conceptualisation

3 Analysis of methodological approaches

The tools prepared for these three steps are given below:

### 1. Tool for analysing values & principles

| | Values & Principles | Values & Principles Relevant to the Research Yes/ No (a) | If Relevant: Present/Limited/ Absent (b) | Observations (on how the incorporated values have been used in the research) (c) |
|---|---|---|---|---|
| **Health System goals** | Sustainability | | | |
| | Equity | | | |
| | Context appropriateness | | | |
| | People-centredness | | | |
| **Health System functioning** | Effectiveness | | | |
| | Safe, Rational, Ethical Practice | | | |
| | Appropriate Technology | | | |
| | Technical efficiency | | | |
| | Affordability | | | |
| | Ecological sensitivity | | | |
| | Dignity in care | | | |

| | Values & Principles | Values & Principles Relevant to the Research Yes/ No (a) | If Relevant: Present/Limited/Absent (b) | Observations (on how the incorporated values have been used in the research) (c) |
|---|---|---|---|---|
| | Self-reliance & Autonomy | | | |
| | Empowerment | | | |
| | Trust & transparency | | | |
| | Accountability & Responsiveness | | | |
| | Decentralisation (dialogic, deliberative, democratic) | | | |
| | Others | | | |

- The reviewer will have to first decide on the values and principles of the health system relevant to the specific research (Column a) based on the research purpose, objectives, and questions. Those listed in the tool were identified from the review based on the lens of the PHC approach with a CAS perspective applied to the whole health system (Fig 1).

- It is important to note that not every HSR must address all the values and principles listed above. By presenting a reasonably comprehensive set of values and principles relevant to Health Systems in the HSRF this step should help the reviewer note down the values present in the research along with their observations, and the researcher at design stage can consciously reflect on the values and principles to adopt. An option of Others is included for the researcher or reviewer adding anything else they think relevant and not covered in this list.

- Then, assess whether they are present, partially present or absent (Column b) in the study under review, further noting any observations they may have about them (Column c).

  The reviewer can refer to the standard definitions of values and principles provided in the glossary (or their references) and note observations about how they are used in the research under review.

- This step will help the reviewer assess whether the incorporated values and their application within the research are appropriate to the research purpose and objectives for assessment in Component B of the DHSRF.

## 2. Tool for analysing Health System Conceptualisation

| Conceptual framework (name or briefly describe the framework) | | | | |
|---|---|---|---|---|
| | | Components Relevant to the Research (note those from options given in the previous column) Yes/No (a) | If Relevant: Present/Limited/Absent (b) | Observations (c) |
| Health System Conceptualisation | Boundary<br>• Addressing Ecological & Social Determinants of Health<br>• Formal Health Service System<br>• Informal Health Care | | | |
| | Subsystems of formal health service system (Structures and processes)<br>• Arrangements to address Ecological and Social Determinants<br>• Service delivery<br>• Health Workforce<br>• Information/knowledge<br>• Health technologies<br>• Finance<br>• Leadership & Governance<br>• Community engagement<br>• Others | | | |

| Conceptual framework (name or briefly describe the framework) | | | Components Relevant to the Research (note those from options given in the previous column)  Yes/No (a) | If Relevant: Present/ Limited/Absent (b) | Observations (c) |
|---|---|---|---|---|---|
| | Informal Health Services (Structures and processes)<br>• Informal providers of allopathy<br>• Traditional caregivers of codified traditional systems<br>• Traditional caregivers of non-codified systems (Local Health Traditions) | | | | |
| | Health System Vantage Point<br>• Top-down/Institutional<br>• Bottom-up/Community centred<br>• Combined | | | | |
| | Dynamic Elements of Health System and its context | Ecosystems<br>• Interrelationships between living organisms and their surroundings influencing health | | | |
| | | Socio-political contexts of health and health care<br>• Combined social and political factors that influence people's health<br>• Socio-political factors that influence health care | | | |
| | | Meaning systems of health and health care<br>• Collective and individual health-related worldviews, experiences and perceptions | | | |
| | | Informal social arrangements & Community practices for health<br>• Health seeking behaviour<br>• Self-care<br>• Household-level care<br>• Societal arrangements (e.g., for leisure, exercise, sports)<br>• Community practices (e.g., seasonal foods; maternal & childcare)<br>• Emergent properties<br>• Others | | | |
| | Relationship of the elements across the system | | | | |
| | Theory of change | | | | |

- Similar to the previous tool, this table must be filled out as per the study's attributes under review (as depicted in Fig 2). All the attributes listed above do not apply to every research study on health systems.

- The reviewer will have to first decide on the components relevant to the specific research (Column a) based on the research purpose, objectives, questions and relevant values and principles and then assess whether they are present, partially present or absent (Column b) in the study under review, further noting down any observations they may have about them (Column c).

- This begins with an assessment of the boundary of the health system in the HSR under review. The following two sections would involve enlisting the components of either the formal or informal health services or both, whichever is being researched. It is important to note here that the observations should include the structure of the health services and the processes that shape their functioning and effectiveness.

- Health System Vantage point, the next section, delineates whether the conceptualisation of the health system is bottom-up, top-down or combined (See Fig 2B). A bottom-up vantage point will involve incorporating people's perceptions, practices, and their underlying reasoning about the aspects of the health system under study. On the other hand, a top-down vantage point primarily involves institutional dimensions and perspectives of health policy, planning, and service delivery. A combined vantage point would integrate both these perspectives in conceptualising the health system and its functioning. For further information, please see the glossary in the S3 File.

- The Dynamic elements of a health system and its context that shape health system structure and processes have been categorised under four heads. Any specific study is expected to have elements in different combinations as relevant to its research questions. 'Ecosystem' is the first element that involves environmental dimensions that affect health. It looks into the interactions between the natural and infrastructural environment in which people live and work or are exposed to, and their health status. This is followed by 'socio-political contexts', that is, the socio-economic conditions and political factors that influence health and health care. 'Meaning systems of health and health care' involve health-related worldviews, experiences and perceptions of people (various institutional actors and/or service users, social sections and communities). 'Informal social arrangements and community practices of health' suggests incorporating the informal dimensions of health care, such as health-promoting or preserving practices of people by themselves, like exercising or following diets for particular health benefits, to social avenues like sports, community activities that determine health, or use of home remedies. The last two elements, Meaning systems and Informal social arrangements, encompass lay-people's cultural context, perceptions/knowledge, and patterns of practice/health-seeking behaviours.

- After enlisting the above and understanding how and in what ways they have been conceptualised in the research, it is crucial to see if these components and elements are considered in silos or in relation to each other. The systems approach in HSR necessitates drawing interrelationships across these elements of the system to understand the system's functioning better, as these lend health systems dynamicity and emergent properties. For instance, to understand the effectiveness of formal health services, an assessment of the status and functioning of the formal health system will only present part of the story unless its relationship with people's health-seeking behaviour is examined. Deviations in actual implementation from the planned programme or service system may reflect the perceptions of the implementers at various institutional levels. The implementers and the user communities could respond in diverse ways to the service design, indicating 'emergent behaviour' within the system. A historical analysis is essential to understand the evolution of various structural and functional attributes of the health system and the 'path dependency' generated that may be influencing the present.

- Any HSR investigates factors and mechanisms within health systems that may bring about changes or, in the case of evaluation, examines such factors and mechanisms to understand whether they have contributed to the desired outcomes. The theory of change is the hypothesis about how the factors and mechanisms are conceptually linked to the expected change. This may or may not be explicitly stated in the HSR. When unstated, it can usually be discerned from the research objectives and their operationalisation. For instance, India's National Rural Health Mission's (NRHM) objective of improvement of rural health care and strategy of architectural correction indicates that improvement of rural health infrastructure and service provisioning along with making a community connect with the Accredited Social Health Activist (ASHA) and communitisation is expected to bring about the desired outcome of improvement of rural health care, its greater utilisation, and its health status outcomes. This is the implicit theory of change and any HSR evaluating NRHM will have to assess this theory of change to see whether this has been effective, or not. Also, have there been unintended or unanticipated developments?

## 3. Tool for analysing Methodological Approaches

| Methodological Approaches | | Approaches Relevant to the Research (note the relevant dimensions) (a) | If Relevant: Present/Limited/Absent (b) | Observations (c) |
|---|---|---|---|---|
| Analytical approaches | Epidemiological approach (Health profile and determinants) | | | |
| | Historical analysis | | | |
| | Social Science approaches<br>• Political Science approach (e.g., Political economy of health)<br>• Sociological approach<br>• Social geographic approach<br>• Economic analysis of health care<br>• Social Anthropological approach<br>• Knowledge pluralism and politics of knowledge<br>• Analysis of social stratification/power/hierarchy<br>• Others | | | |
| | Management Science approaches | | | |
| | Ecosystem approaches | | | |
| Operational approaches*<br>• Intervention study design<br>• Non-Intervention study design | | | | |
| Nature of disciplinary interaction<br>• Unidisciplinary<br>• Multidisciplinary<br>• Interdisciplinary<br>• Transdisciplinary | | | | |
| Ethical Considerations of the Study (Consent, Confidentiality, Conflict of Interest) | | | | |
| Quality Assessment of Research Operationalisation** | | | | |

Note:

* Operational approaches will have to be further detailed depending on the methodology applied.

** Any quality assessment tool deemed relevant for the research can be used.

- Similar to the other two tools, this table must be filled for various methodological features (as depicted in Fig 3) as appropriate to the study under review. The glossary in the S3 File describes each term used in the DHSRF.

- The reviewer will have to decide on the methodological approaches relevant to the specific research (Column a) based on the research purpose, objectives, questions, values and principles and health systems conceptualisation, assess whether they are present, partially present or absent (Column b) in the study under review, and note down any specific observations (Column c)

- The Analytical approaches section includes the major disciplines that contribute to HSR. In addition to listing the disciplinary approach used in the study, the reviewer can make further observations about how such an approach was used. For instance, a historical analysis can mean just a background overview or a more in-depth analysis of the historical trajectory of the health system in that particular context and how that may continue to shape the health system.

- The section on operational approaches will detail the study's operationalisation, whether it is an intervention or no-intervention study, qualitative, quantitative, or mixed methods research, and observe its fit with the research questions.

- The following section is about discerning the cross-disciplinary interactions used in the study. Details of each are included in the glossary.

- The final section in this step is assigning the research paradigm. Summary attributes of each paradigm are presented in Table 2. Please refer to the relevant section in the glossary for further details.

Application of the DHSRF for review of HSR studies can be seen illustratively in S4 File and its application for designing studies can be seen in S5 File.

It is evident that these analytical steps presented in the tools above are not just means to a review of research. They provide an important window into the black box of research while shedding light on the underlying research paradigms that inform research. Unveiling the inner workings of HSR and rendering it accessible for assessment is likely to support better scientific development of the research. Conscious decisions on the selection of research paradigms and their articulation, theoretical conceptualisation, and methodological approaches will help facilitate a self-critical and rigorous approach to designing a study and facilitate more open conversations across different streams involved in HSR, most of which at present are fragmented and confined to disciplinary silos. Hence, the DHSRF is not just suitable for reviewing HSR but can also act as a reference tool for designing HSR. For examples of review done using the DHSRF please see the S4 File.

## Steps of application of the HSRF for designing an HSR study

- Step 1: Refer to the Values and Principles tool to decide the values and principles that would be appropriate for the research objective/s.

- Step 2: Delineate the research questions keeping in mind the different underlying values.

- Step 3: Apply the three main research paradigms, Positivist, Critical Realist and Critical Holist, for a preliminary assessment of the conceptual-methodological approaches required to address the research questions. Based on what is most relevant for the research objective and feasibility, decide on the research paradigm or their combination for the study.

**Table 2. The Research Paradigms identified in HSR Studies and their summary attributes.**

| Name of Research Paradigm | Tendencies & Attributes |
|---|---|
| Conventional Comprehensive Public Health (Pragmatist) | Epidemiological and problem-oriented approach; Boundary placed as relevant to problem-solving, Minimally defined linkages with the social processes and contexts, Primarily quantitative methods informed by tacit contextual knowledge defining linkages with social dimensions and processes, Tends to be Top-down |
| Positivist | Deductive and quantitative in methodology; Narrow boundary relative to objective, Limited consideration of historicity, contexts and diversity, and dynamic processes; Techno-managerial and Top-down approach |
| Realist closer to positivist | Narrow boundary, context-mechanism-outcome with minimal attention to context and linkages; mixed methods; Top-down |
| Realist | Context-mechanism-outcome configuration as the guiding framework; Social science theory-driven questions; Boundary placed as relevant to the research questions; Diversity of perceptions of mechanisms are foregrounded; use of mixed methods; In practice tends to exclusively focus on the contemporary, with minimal attention to historical analyses and epidemiology; focus on the context of the formal system with minimal attention to cultural context and none to knowledge pluralism and bottom-up perspective |
| Realist closer to Holist | Wider boundary with greater attention to context-mechanism-outcome configurations; Studies context and linkages with a minimum of one and a maximum of three of the following: Historical analysis, Epidemiological orientation, Knowledge pluralism and Bottom-up system vantage point; mixed methods |
| Critical Holist | Attempts to capture the complexity of the system by drawing from diverse forms and sources of knowledge towards problem-solving; It considers the widest possible configurations of ecological-socio-political and cultural contexts, incorporating epidemiology, historical analysis, knowledge pluralism and the politics of knowledge and bottom-up perspectives. Thereby identifies the critical dimensions relevant to the problem through dialogic and transdisciplinary research. |

- Step 4: Refer to the Tool for Health Systems Conceptualisation, to decide on the relevant boundary, components and elements for the health system/aspect being studied.

- Step 5: Refer to the Tool for Methodological Approaches to decide on the research methodology.

- Step 6: After deciding the above, check for research coherence across these steps before proceeding to implement the actual research.

## How the Indian health system informed the DHSRF

Before concluding, we want to highlight how certain key considerations from our knowledge and experience of the Indian health systems informed the DHSRF.

While we wanted the framework to be flexible enough to be applicable across different contexts with appropriate adaptations, we also made sure that it was sensitive to the immediate context for which it was being developed. Therefore, in addition to making it dialogic, as explained above, the ideas of context specificity, capturing diversity of perspectives and value criticality were paramount to the development of the DHSRF. For instance, our knowledge of the Indian health systems' past and present contexts made us think of including *Informal Health Services (Structures and processes)* in health systems conceptualisation. This was because in India, a significant section of the population resort to informal health care from traditional systems of medicines or local health traditions for their health needs. While there may be conflicting evidence and opinions about their effectiveness, including them in the DHSRF allows us to capture the diverse perspectives on health seen in the actual practice of health. Therefore, this allows for a much more holistic picture of the health system and health-seeking behaviours. In a similar vein, we have included *Informal social arrangements & Community practices for health* as these diverse health-producing practices that have existed for many generations in our society, or in the homestead, or those that have been newly inculcated often get neglected at the cost of capturing the most visible, the formal health system, public or private. This also led to addressing the issues of CAS; where the knowledge of our history and context informed the DHSRF, making it more sensitive than many other existing conceptualisations to path dependency, emergent behaviours, and feedback loops. [11] Most importantly, we understand that researchers come from different training and value positions that shape their priorities in their health systems research. Instead of being judgemental about any of these positions, the DHSRF allows the researcher to be upfront about their value positions that inform their choices. This step of value criticality not only allows for a better understanding of the research process but also galvanises more informed debate and discussion.

Thus, the DHSRF is not just a tick-box exercise. It is intended to prod the researcher or the reviewer to think more deeply about what all goes into the conceptualisation of health systems research, and more importantly, what informs those choices. The DHSRF is flexible enough for the researcher to choose what applies and what does not for their research context and also to adapt it as they see fit. It attempts to make everything explicit and over the board for constructive dialogues in advancing the field of health systems research and facilitating informed and thoughtful use of research outcomes by policy makers.

## Conclusion

While there has been significant upsurge in HSR in the last few decades, there has been a rise in fragmentation of the field too with little dialogue across different positions. It is to some extent to be expected, given how the development in HSR has been in response to previous work or what it was considered to be lacking more precisely. Given the vastness and complexity of the field, we feel this diversity is not only inevitable but also a crucial strength of the field. But, for diversity to really matter in highlighting neglected voices and perspectives for more comprehensive, sustainable and just health systems, it needs to rise beyond a cacophony to create meaningful conversations with one another. Conversations can only happen through a common language and our hope is that the Dialogic Health Systems Research Framework (DHSRF) can be a tool to facilitate such conversations.

By making transparent the inner workings of any HSR, DHSRF hopes to create open dialogues that will help health system researchers learn from each other and broaden the field in the process. The transparency will also likely make the evidence base of HSR more robust as everything, including normative choices will be explicitly stated for anyone to assess. We see three primary uses of the framework in practice -

1. To serve as a reference guide for the researchers before commencing on HSR and in the process of designing their study to assess conceptual, methodological and normative choices.

2. To serve as a guide for reviewers to assess the robustness of theoretical and methodological conceptualisation of an HSR study relative to its objective. This can help in assessing the strength of research proposals for appropriate support and the strength of the evidence generated as outcome of the research.

3. Based on the strength of the evidence as analysed using the DHSRF, a comprehensive review can be made of different policy choices related to the values and principles desired for the health system. This can potentially help in revising the programme formulation, budget outlays and delivery strategies for better outcomes through mid-term reviews of major programmes as well as in the end evaluation. This would bring rigour to and foster a culture of health systems research led evidence-informed policy making.

But primarily, we hope that the DHSRF will facilitate greater self-criticality, dialogue and collaborations in HSR towards health for all.

## Supporting information

**S1 File. Systematic Narrative Review Methodology & Versions of Health System Research Frameworks (Version 0–3).**
(PDF)

**S2 File. Definition of HSR.**
(PDF)

**S3 File. Glossary.**
(PDF)

**S4 File. Application of the DHSRF to Review Studies: Some Illustrations.**
(PDF)

**S5 File. Illustration of Designing of HSR using the DHSRF.**
(PDF)

## Acknowledgments

We wish to thank the Technical Advisory Group members and all participants of the Researchers' consultation who engaged actively with the draft HSRF that was presented to them and gave very constructive comments and suggestions, based on which we further developed the tool.

## Author contributions

**Conceptualization:** Ritu Priya, Sayan Das, Amitabha Sarkar, Atul Kotwal.

**Data curation:** Ritu Priya, Sayan Das, Liz M. Kuriakose.

**Formal analysis:** Ritu Priya, Sayan Das, Liz M. Kuriakose.

**Funding acquisition:** Atul Kotwal.

**Investigation:** Ritu Priya, Sayan Das, Liz M. Kuriakose, Madhurima Shukla, Amitabha Sarkar.

**Methodology:** Ritu Priya, Sayan Das, Liz M. Kuriakose, Madhurima Shukla, Amitabha Sarkar.

**Project administration:** Ritu Priya, Neha Dumka, Erin Hannah, Nisha Basheer, Atul Kotwal.

**Resources:** Ritu Priya, Neha Dumka.

**Supervision:** Ritu Priya, Neha Dumka, Erin Hannah, Nisha Basheer, Atul Kotwal.

**Validation:** Ritu Priya, Neha Dumka, Erin Hannah, Nisha Basheer, Atul Kotwal.

**Visualization:** Ritu Priya, Sayan Das.

**Writing – original draft:** Ritu Priya, Sayan Das, Liz M. Kuriakose.

**Writing – review & editing:** Ritu Priya, Sayan Das, Liz M. Kuriakose, Madhurima Shukla, Neha Dumka, Erin Hannah, Nisha Basheer, Atul Kotwal.

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
