## [Decision Letter · Decision Letter 0]

16 Apr 2025

PGPH-D-24-03072

The Dialogic Health Systems Research Framework (DHSRF): A tool for facilitating self-criticality, researcher interactions and knowledge management in Health Systems Research & Policy Studies

Dear Dr. Priya,

Thank you for submitting your manuscript to PLOS Global Public Health. After careful consideration, we feel that it has merit but does not fully meet PLOS Global Public Health’s publication criteria as it currently stands. Therefore, we invite you to submit a revised version of the manuscript that addresses the points raised during the review process.

We look forward to receiving your revised manuscript.

Kind regards,

Nafis Faizi, MD, MPH

Academic Editor

Journal Requirements:

1. Please provide additional details regarding participant consent. In the ethics statement in the Methods and online submission information, please ensure that you have specified (1) whether consent was informed and (2) what type you obtained (for instance, written or verbal, and if verbal, how it was documented and witnessed).

2. We ask that a manuscript source file is provided at Revision. Please upload your manuscript file as a .doc, .docx, .rtf or .tex.

Additional Editor Comments (if provided):

Reviewers' comments:

Reviewer's Responses to Questions

**Comments to the Author**

1. Does this manuscript meet PLOS Global Public Health’s publication criteria ? Is the manuscript technically sound, and do the data support the conclusions? The manuscript must describe methodologically and ethically rigorous research with conclusions that are appropriately drawn based on the data presented.

Reviewer #1: Yes

Reviewer #2: Yes

Reviewer #3: Yes

2. Has the statistical analysis been performed appropriately and rigorously?

Reviewer #1: N/A

Reviewer #2: N/A

Reviewer #3: N/A

3. Have the authors made all data underlying the findings in their manuscript fully available (please refer to the Data Availability Statement at the start of the manuscript PDF file)?

Reviewer #1: Yes

Reviewer #2: Yes

Reviewer #3: Yes

4. Is the manuscript presented in an intelligible fashion and written in standard English?

Reviewer #1: Yes

Reviewer #2: Yes

Reviewer #3: Yes

5. Review Comments to the Author

Reviewer #1: Thank you for an opportunity to review this manuscript. I thought the idea of having a research framework particular to HSR maybe useful to people shifting into doing this kind of research. The framework also gives a list of items to think about while doing any research- not just systems research.

It also seems to have merit as a teaching resource.

- As a person, or perhaps the training I come from, I found the tools long and too structured- with too many rows and columns.

- I agree that there is merit in thinking about values in HSR, situating it in a research paradigm etc. But not always. This kind of thinking may suit core academic research ( eg Phd work). But not always when eminent national research bodies ( eg NHSRC) have to produce HSR quickly. HSR also needs to quickly respond to situations, draw from tacit knowledge, and be rooted in practice. Those elements are missing from the framework as of now.

- Also, the complexities of systems thinking are not really present in the framework (only mentioned in the beginning). This could be a limitation.

- It would be nice to incorporate the idea of beginning research from a ‘real problem’ in the field, and seeing if systems thinking and HSR in general as a way to help solve the problem.

In the present form, the framework appears to be overly focussed on situating the research into a set of research paradigms, system elements, and values- which I feel may not always be necessary or relevant to do.

-

But overall, I think this is an interesting piece of work and happy for peers to engage with it once published.

Reviewer #2: Reviewer comments

The paper titled dialogic health systems research framework attempts to put forth a framework for health system researchers that is holistic, value based and methodologically rigorous so that researchers in Health systems and policy research can be self-critical and reflexive on the approaches and the methodology adopted. The paper is too complex to comprehend in its current form as the concepts of health systems itself is contested, and hence calls for an explicit position of health system by the authors before attempting to address a framework for health systems research. However, to achieve the intended objectives, the paper deserves the intervention of the authors, aiming to provide more substantial elements for health systems and the varied purposes of health systems research.

Following are my suggestions to improve the paper:

The clarity on the concept of health systems is lacking in the paper as there are instances in the paper where HSR and HPSR are interchangeably used (p.5) while referring to AHPSR, which emerged in a different context. Whether HSR and HPSR are seen as similar by the authors need to be stated explicitly for better clarity. Those details in the supplementary material (S 2) about health systems need to be necessarily brought into the main paper for better clarity by reducing the introduction section.

Second aspect is about the framework, whether proposing a uniform research framework for studying an event/ phenomenon, here the health systems, can also constrains the possibilities of research. By proposing a single research framework, here DHSRF, for studying an event/ phenomenon, it might bring in more rigidity in the idea of research itself. Rather for every research, it is important to make its epistemological and ontological assumptions explicit not only for ensuring rigour and validity of the methodology but also to understand the standpoint and purpose of the researcher. Unfortunately, that is missing in several of the current research and is true with the field of HSR as well. The three step tools of analysis mentioned in the framework resonates with the epistemological foundations that guide any research, namely, values and principles underlying the concepts of health systems and the actual conceptualisation, which is about the ontological assumptions and finally the methodology is about the paradigm. In the values and principles, it is the primary health care approach and the eco-social frameworks that is proposed, which is acceptable from a public health perspective. These three step tools can be organised better by separating ontological aspects of health systems from that of theoretical and methodological approaches and so on.

Finally, every research does have diverse standpoints in its approach to the event/ phenomenon. Hence, the disciplinary standpoint for health systems research could be better justified if the public health disciplinary standpoint is adopted as central to health systems research. Health systems strengthening, people centredness, equity and sustainability then become characteristics of a public health disciplinary standpoint and can be proposed as the major purpose of all kinds of Health systems research. HSR historically was developed keeping the public health value system as central. It is the neoliberal context of individualism that has led to the domination of market values of economics and techno-managerial, and reductionism as the focus of HSR at the cost of public health standpoint. There is a need to regain it and can be demonstrated with the analysis of the existing studies on HSR as attempted in the analysis of the research papers in this article. Several of the content in the supplementary files need to be included in the paper for better clarity.

Reviewer #3: The authors have done a great and commendable work in creating this framework that consolidates the field of Health Systems Research (HSR). This framework will help to fill the gap in the HSR and bring researchers closer to speaking a more unified language. The interdisciplinary nature of this field is both an advantage and disadvantage. I therefore truly thank the researchers for the effort they put into this work and coming up with this comprehensive framework based on HRS work in India. The paper is well written and interesting to read. Well done!

Here are a few comments for the authors’ consideration.

1. Methodology section – It would be great if the authors indicated the overarching methodological approach for this process presented in this paper.

2. In lines 236-241, the authors indicate that they found out that ‘similar research objectives and questions, the adoption of different paradigms lead to different conceptualisation of the problem…’ this to me sound rather upside down because usually conceptualisation of the problem comes before formulating research questions and questions and research objectives. I still believe the statement from the AHPSR methodology reader makes sense. I would consider rewording the above statement or removing it as it is rather confusing.

3. The concept of ecosystem is not clearly articulated like other concepts. It would be useful for the researchers to shed more light on this.

4. There is a small typo on line 536. Instead on ‘This’ , its currently ‘his’.

6. PLOS authors have the option to publish the peer review history of their article (what does this mean? ). If published, this will include your full peer review and any attached files.

**Do you want your identity to be public for this peer review?** For information about this choice, including consent withdrawal, please see our Privacy Policy .

Reviewer #1: No

Reviewer #2: No

Reviewer #3: No

---

## [Editor Report · Decision Letter 1]

10 Jul 2025

The Dialogic Health Systems Research Framework (DHSRF): A tool for facilitating self-criticality, researcher interactions and knowledge management in Health Systems Research & Policy Studies

PGPH-D-24-03072R1

Dear Dr. Priya,

We are pleased to inform you that your manuscript 'The Dialogic Health Systems Research Framework (DHSRF): A tool for facilitating self-criticality, researcher interactions and knowledge management in Health Systems Research & Policy Studies' has been provisionally accepted for publication in PLOS Global Public Health.

Best regards,

Nafis Faizi, MD, MPH

Academic Editor